# Heparan sulfate proteoglycans present PCSK9 to the LDL receptor

Camilla Gustafsen [1], Ditte Olsen[1], Joachim Vilstrup [2], Signe Lund[1], Anika Reinhardt[3], Niels Wellner[1], Torben Larsen[4], Christian B.F. Andersen[1], Kathrin Weyer[1], Jin-ping Li[5], Peter H. Seeberger[6], Søren Thirup [2], Peder Madsen[1] & Simon Glerup[1]

Coronary artery disease is the main cause of death worldwide and accelerated by increased plasma levels of cholesterol-rich low-density lipoprotein particles (LDL). Circulating PCSK9 contributes to coronary artery disease by inducing lysosomal degradation of the LDL receptor (LDLR) in the liver and thereby reducing LDL clearance. Here, we show that liver heparan sulfate proteoglycans are PCSK9 receptors and essential for PCSK9-induced LDLR degradation. The heparan sulfate-binding site is located in the PCSK9 prodomain and formed by surface-exposed basic residues interacting with trisulfated heparan sulfate disaccharide repeats. Accordingly, heparan sulfate mimetics and monoclonal antibodies directed against the heparan sulfate-binding site are potent PCSK9 inhibitors. We propose that heparan sulfate proteoglycans lining the hepatocyte surface capture PCSK9 and facilitates subsequent PCSK9:LDLR complex formation. Our findings provide new insights into LDL biology and show that targeting PCSK9 using heparan sulfate mimetics is a potential therapeutic strategy in coronary artery disease.

---

[1] Department of Biomedicine, Aarhus University, Ole Worms Allé 3, 8000 Aarhus, Denmark. [2] Department of Molecular Biology and Genetics, Aarhus University, Gustav Wieds Vej 10, 8000 Aarhus, Denmark. [3] Scienion AG, Volmerstrasse 7b, 12489 Berlin, Germany. [4] Department of Animal Science, Aarhus University, Blichers Allé 20, 8830 Tjele, Denmark. [5] Department of Medical Biochemistry and Microbiology, University of Uppsala, Husarg. 3, 75237 Uppsala, Sweden. [6] Department of Biomolecular Systems, Max Planck Institute for Colloids and Interfaces, Am Mühlenberg 1 OT Golm, 14476 Potsdam, Germany. Correspondence and requests for materials should be addressed to C.G. (email: gustafsen@biomed.au.dk) or to S.G. (email: glerup@biomed.au.dk)

ncreased level of plasma low-density lipoprotein (LDL) cholesterol is considered a key predictor for the development of coronary artery disease (CAD), which is the main cause of death in the world. The primary choice of medication is statins, and these are among the most commonly prescribed drugs worldwide. Statins inhibit endogenous cholesterol synthesis and concomitantly increase the expression of the low-density lipoprotein receptor (LDLR) in hepatocytes[1], resulting in increased uptake of LDL cholesterol particles from the circulation by LDLR-mediated endocytosis. LDL is subsequently degraded in lysosomes and cholesterol is recovered for use in the hepatocyte or conversion to bile acids while LDLR recycles to the cell surface. Unfortunately, a considerable number of patients show insufficient response and do not reach the desired levels in plasma LDL cholesterol[2]. Statins also increase the expression and secretion of proprotein convertase subtilisin/kexin type 9 (PCSK9) in hepatocytes[3, 4]. PCSK9 is structurally related to the proprotein convertases but proteolytically inactive due to tight association between the prodomain and the catalytic domain[5]. PCSK9 binds LDLR on the surface of hepatocytes and triggers its degradation in lysosomes thereby counteracting the beneficial effects of statins at the posttranslational level. Accordingly, inhibition of the PCSK9:LDLR interaction efficiently reduces plasma LDL cholesterol, and the first two humanized antibodies blocking the LDLR-binding site in PCSK9 recently received final clinical approval for treating patients suffering from hypercholesterolemia[6–8]. However, it remains a mystery how the soluble monomeric protein PCSK9 dramatically can change the cellular trafficking route of the single-pass transmembrane receptor LDLR from recycling to lysosomal degradation[9, 10]. Furthermore, the PCSK9:LDLR binding constant is in the range of 170–628 nM[11, 12] while the PCSK9 plasma concentration is around 6 nM[13], rendering it unlikely that circulating PCSK9 binds LDLR directly at normal physiological concentrations. In addition, PCSK9 targets LDLR in the liver but not in, e.g., steroid hormone-producing tissues, which also express high levels of LDLR, suggesting the requirement of a liver-specific co-receptor[5, 14, 15].

The hepatocyte surface is covered with heparan sulfate proteoglycans (HSPG) that are known to play important physiological roles in several aspects of lipoprotein metabolism including endocytosis of bound ligands[16]. Heparan sulfate is composed of repeating disaccharide units consisting of glucuronic acid or iduronic acid (IdoA), which can be O-sulfated, and N-acetyl glucosamine (GlcN), which can be both O-sulfated and N-sulfated, in an apparently specific and cell type-dependent manner[17]. Heparin is a highly sulfated variant of heparan sulfate obtained as a heterogeneous specie typically from porcine entrails or equine lungs, and is the biopharmaceutical produced at the largest scale worldwide due to its potent anticoagulant activity[17]. In the present study, we observed that the amino acid sequence of the PCSK9 prodomain contains a cluster of basic residues in agreement with consensus sequences for interaction with HSPG[17, 18]. We further find that these are essential for PCSK9 activity in vitro and in vivo and propose a model in which HSPG capture and present PCSK9 to LDLR at the hepatocyte surface. Accordingly, antibodies directed against the HSPG-binding site, heparin or heparan sulfate mimetics are PCSK9 inhibitors and may serve as a potential treatment for CAD.

## Results

**PCSK9 binds HSPG.** We examined the electrostatic surface of PCSK9 and identified a putative heparin-binding site composed of six surface-exposed basic residues located in the PCSK9 prodomain. The binding site is formed by arginine (R) residues at position 93, 96, 97, 104, and 105 and histidine (H) at position 139, which docked with sulfate groups of a heparin penta-saccharide (SANORG) (Fig. 1a, b). The site is found opposite to the LDLR binding surface located in the inactive catalytic domain of PCSK9 (Supplementary Fig. 1a). Placing heparin onto a co-crystal structure of PCSK9 in complex with LDLR further suggested that heparan sulfate binding allows subsequent PCSK9:LDLR complex formation (Supplementary Fig. 1b). We next treated human hepatocyte-derived HepG2 cells stably expressing PCSK9 with heparinase I, which cleaves heparan sulfate GAG chains between GlcN and IdoA, thereby removing cell surface heparan sulfate. Indeed, treatment resulted in a marked reduction in the intensity of surface PCSK9 staining (Fig. 1c). No decrease was observed upon treatment with chondroitinase (Supplementary Fig. 2a). To exclude that PCSK9 cell surface binding was inhibited in an unspecific manner by heparinase-generated heparan sulfate fragments, we first treated cells with heparinase then thoroughly washed the cells, and subsequently incubated them with purified PCSK9 on ice to prevent endocytosis. In this experiment, heparinase treatment completely prevented PCSK9 cell surface binding, suggesting that HSPG are indeed PCSK9 receptors (Supplementary Fig. 2b). To study the direct interaction between PCSK9 and heparan sulfate GAG chains, we employed affinity chromatography using Sepharose beads covalently coupled with heparin. Purified PCSK9 was retained on the heparin column and eluted at a NaCl concentration of approximately 500 mM (Fig. 1d), confirming the interaction with heparin. We further found that endogenous PCSK9 from conditioned media of HepG2 cells also bound heparin (Supplementary Fig. 3a) and showed similar elution profile as that of ApoE, a well-established HSPG-binding protein[19]. To determine if the predicted HSPG-binding motif in PCSK9 is responsible for heparin binding, we cloned and expressed PCSK9 variants in which combinations of the six amino acids were substituted for alanine (A) (Supplementary Fig. 3b). All PCSK9 mutants were processed and secreted in a similar manner from transfected Chinese Hamster Ovary (CHO) cells (Supplementary Fig. 3c), indicating correct folding as this is prerequisite for processing of proPCSK9 and subsequent exit from the endoplasmic reticulum[20]. Substitution of combinations of the six basic amino acids predicted in Fig. 1a, b, including substitution of R93 to alanine (mut 1) or cysteine (R93C), prevented binding (Fig. 1e and Supplementary Fig. 3d), demonstrating that the site is responsible for the interaction with heparin. Of note, PCSK9 R93C is a naturally occurring loss-of-function variant[21, 22]. In contrast, substitution of other surface-exposed basic residues positioned outside the predicted HSPG-binding motif in PCSK9 (R165 and R167, mut 6) did not have any major effect on heparin binding (Supplementary Fig. 3d, e).

**HSPG binding is required for PCSK9-induced LDLR degradation.** We incubated HepG2 cells for 18 h with wild-type (WT) PCSK9 or a PCSK9 basic residue mutant lacking all six basic residues (mut 5), and subsequently analyzed the LDLR levels by Western blotting. We found that PCSK9 WT was approximately twice as effective in inducing LDLR degradation compared to the PCSK9 mut 5 (Fig. 1f, g), suggesting that HSPG binding is critical in PCSK9-induced LDLR degradation. Importantly, purified PCSK9 WT and PCSK9 mut 5 showed similar binding to the immobilized extracellular domain of LDLR as determined using Biacore (Supplementary Fig. 4a, b). We speculated that exogenously added heparin could compete with cell surface HSPG for the binding of PCSK9 and thereby prevent PCSK9:LDLR complex formation. Using proximity ligation assay (PLA), abundant surface clusters of PCSK9:LDLR complexes were observed in non-permeabilized HepG2 cells (Fig. 1h). These were markedly reduced both in numbers and intensity upon incubation with

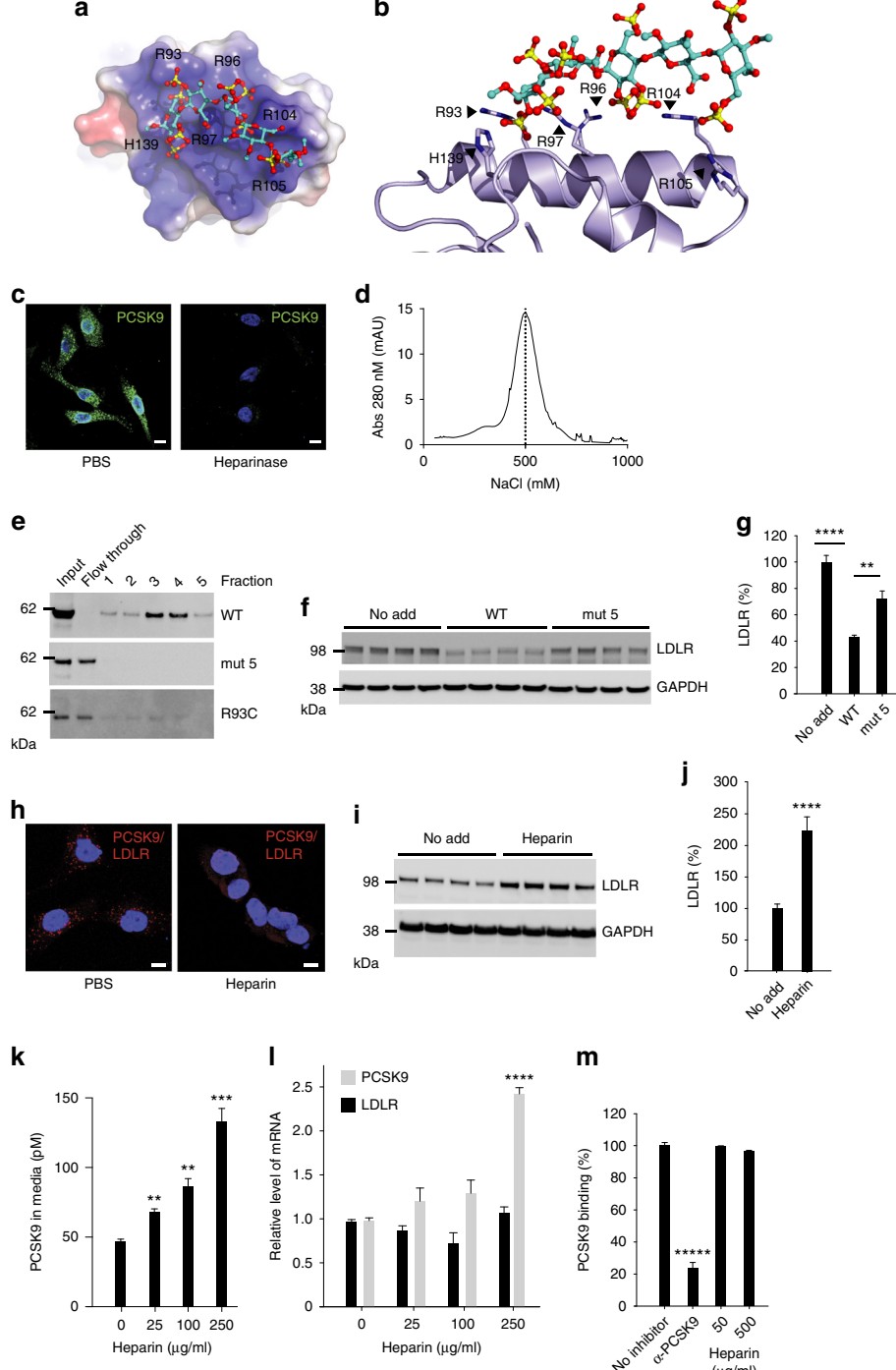

**Fig. 1** PCSK9 cell surface binding and activity depend on HSPG. **a** A heparin pentasaccharide (SANORG, sticks) (PDB ID 1E03)[61] docked at the electrostatic surface (*red* negative; *blue* positive) of the HSPG-binding site in PCSK9 (PDB ID 3H42)[26] with positively charged amino acids (R: arginine, H: histidine) indicated. **b** Superposition of SANORG with PCSK9 (*ribbon*). **c** Non-permeabilized HepG2 cells-expressing PCSK9 (*green*) after treatment with or without heparinase I. Nuclei were stained with Hoechst (*blue*). The experiment was repeated three times with similar results. **d** PCSK9 binding to heparin was analyzed by affinity chromatography. **e** PCSK9 mutant with all six amino acids highlighted in **a**, **b** substituted for alanines (mut 5) showed complete loss of heparin binding. Substitution of R93 alone for cysteine (R93C) also resulted in complete loss of heparin binding. **f, g** HepG2 cells incubated 18 h with WT PCSK9 and PCSK9 mut 5 (10 nM of each, $n = 4$). **h** PLA analysis of non-permeabilized HepG2 cells showed that co-localization of LDLR and PCSK9 was markedly reduced upon incubation of cells with heparin (500 μg/ml). The experiment was repeated three times with similar results. **i, j** LDLR levels in HepG2 cells following incubation with heparin (500 μg/ml, 18 h, $n = 4$). **k** Heparin treatment resulted in accumulation of PCSK9 in the culture supernatant ($n = 3$). **l** Effect of heparin on LDLR and PCSK9 mRNA levels. **m** Heparin (50 and 500 μg/ml) had no effect on the interaction between PCSK9 and LDLR in a cell-free assay. A PCSK9 antibody (αPCSK9, 5 nM) directed against the LDLR-binding site is included as positive control ($n = 3$). *Bar graphs* show mean values ($n$ as indicated) with s.e.m. error bars. Statistical significance was evaluated using a two-tailed Student's *t*-test. *Scale bars* are 10 μm. Supplementary Fig. 10 shows uncropped gel images

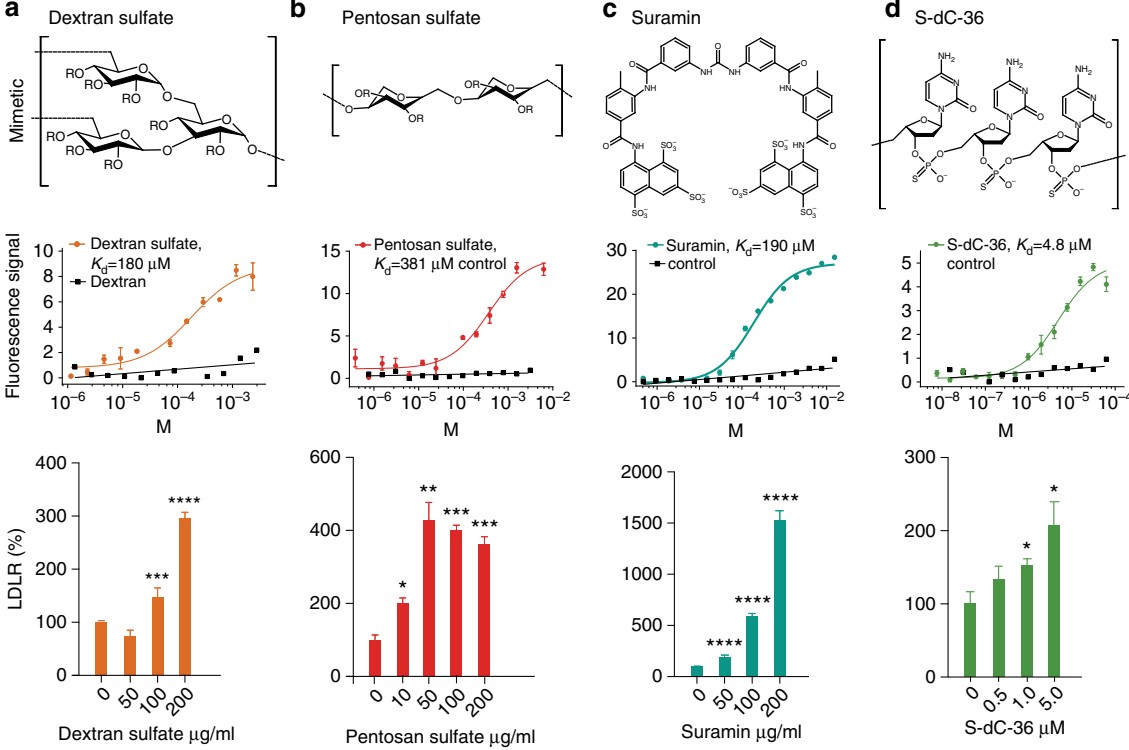

**Fig. 2** Heparin mimetics are potent PCSK9 inhibitors. Structure of the heparin mimetics dextran sulfate (**a**), pentosan sulfate (**b**), suramin (**c**), and the phosphorothioate oligonucleotide S-dC-36 (**d**) (R = $SO_3^-$ or H). Binding curves and the dose-dependent effects on LDLR levels in HepG2 cells ($n = 3–5$, *bar graphs* show mean values with s.e.m. error bars) are shown below each mimetic structure. Statistical significance was evaluated using a two-tailed Student's *t*-test. Summary of binding constants and representative LDLR Western blots are shown in Supplementary Fig. 5d, e

heparin (50 U/ml corresponding to approximately 500 µg/ml), suggesting that PCSK9 binding to HSPG is instrumental in the subsequent complex formation with cell surface LDLR. We found that incubation of HepG2 cells with heparin (500 µg/ml for 18 h) resulted in two to 3-fold higher level of LDLR protein (Fig. 1i, j). A similar effect was observed with two therapeutic preparations of low-molecular weight heparins (LMWH) as well as with heparan sulfate but not with chondroitin sulfate (Supplementary Fig. 5a, b). The increase in cellular LDLR in heparin-treated HepG2 cells was dose-dependent and accompanied by a marked increase in PCSK9 in the medium as determined using a PCSK9-specific enzyme-linked immunosorbent assay (ELISA) (Fig. 1k). The increase in cellular LDLR and extracellular PCSK9 induced by heparin occurred at the post translational level as we observed no change in mRNA, except for PCSK9 at the highest heparin concentration (Fig. 1l). Nevertheless, 250 µg/ml heparin effectively antagonized its activity as evident from approximately 2.5-fold increase in LDLR protein (Supplementary Fig. 5c) despite the 2-fold increase in PCSK9 mRNA. Importantly, heparin did not affect the direct binding of PCSK9 to immobilized LDLR in a cell-free PCSK9 (biotinylated)-LDLR binding assay (Fig. 1m).

**Heparin mimetics are PCSK9 inhibitors.** Several molecules mimicking the structure of heparin have been developed, seven of which are currently in clinical use for a number of indications. These are denoted heparin/heparan sulfate mimetics and belong to diverse chemical classes, including various oligosaccharides, oligonucleotides, and naphthalene derivatives[23]. We tested a subset of mimetics for their ability to bind PCSK9 and increase LDLR in HepG2 cells (Fig. 2a–d and Supplementary Fig. 5d, e). The sulfated oligosaccharides dextran sulfate and pentosan sulfate both bound directly to PCSK9 with affinity constants of 180 and 381 µM, respectively, as estimated using Microscale Thermophoresis (MST), and resulted in dose-dependent increase in cellular LDLR, reaching a plateau of around 400% compared to control. The interaction was dependent on the presence of sulfate groups as non-sulfated dextran showed no affinity for PCSK9. The sulfated naphthalene derivative suramin, an antiparasitic drug used against African sleeping sickness, resulted in up to 15-fold increase in LDLR and concomitantly increased uptake of LDL in HepG2 cells (Supplementary Fig. 5f). Phosphorothioate oligonucleotides are highly anionic and known to interact with heparin-binding proteins[24, 25]. We therefore tested a modified 36-mer phosphorothioate oligodeoxycytidine (S-dC-36) and found that it bound PCSK9 with a $K_D$ of 4.8 µM and showed inhibitory effects at this concentration range. Thus, it appears that a number of negatively charged compounds display inhibitory activity toward PCSK9, and to further understand the structure–activity relationship crystallographic data on a PCSK9: heparin mimetic complex would be desirable. We noted that dextran sulfate was a constituent of the crystallization buffer for the crystal structure of PCSK9 in complex with a Fab fragment of evolocumab[26]. In the PDB data set, we discovered electron density surrounding the α-helix constituting the HSPG-binding site, which was 11-fold above the background level, and had not been accounted for in the published structure. We solved the remaining part of the structure and found it to encompass a dextran sulfate disaccharide interacting with R93, R97, R104, and H139 of PCSK9 (Fig. 3a–c, Supplementary Fig. 6 and Supplementary Table 1). Taken together, these data suggest that the number and positioning of negatively charged functional groups of sulfated sugars and mimetics are critical determinants for their binding and activity toward PCSK9.

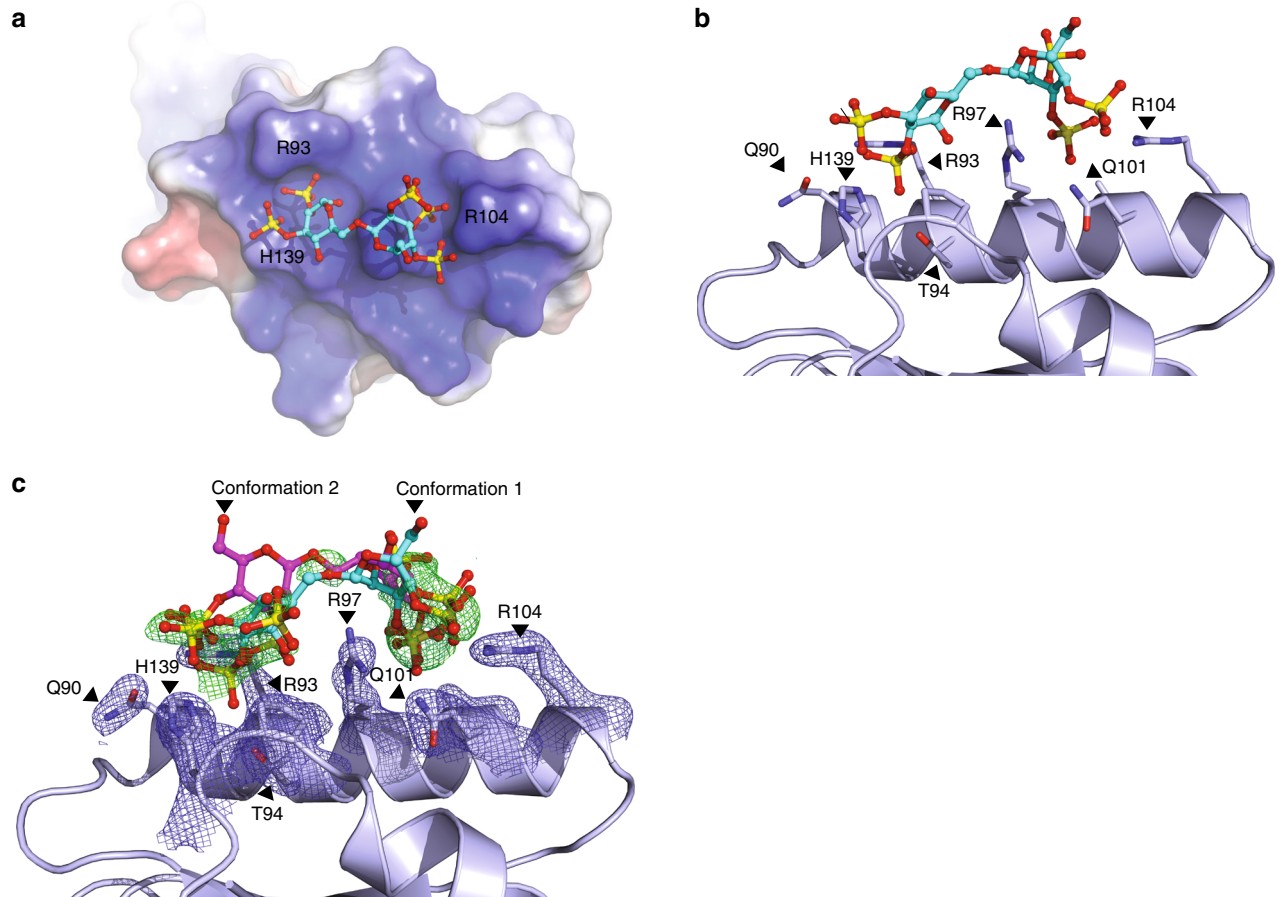

**Fig. 3** Structure of PCSK9 in complex with a dextran sulfate disaccharide. **a** Structure of dextran sulfate disaccharide on the electrostatic surface (*blue*: positive; *red*: negative) of the pro-domain of PCSK9 (PDB ID: 5OCA) with amino acids indicated (H: histidine and R: arginine). **b** Side view of the binding site containing conformation 1 of dextran sulfate with interacting residues indicated (Q: glutamine, T: threonine). **c** Simulated annealing Fc-Fo composite omit map (positive density show in *green* and contoured at $3\sigma$) of dextran sulfate in two alternative conformations (confirmations 1 (*cyan*) and 2 (*magenta*)). Interacting residues of PCSK9 are indicated and show together with a simulated annealing 2Fc-Fo composite omit map (*blue* and contoured at $1\sigma$)

**PCSK9 binds trisulfated heparan sulfate disaccharide repeats.** The extracellular matrix and plasma membrane contain a wide variety of negatively charged glycans such as HSPG and the composition is highly tissue and cell type-dependent. To dissect the specificity and selectivity of the interaction of PCSK9 with negatively charged sugars, we analyzed PCSK9 binding to a glycan microarray consisting of immobilized synthetic extracellular matrix glycans, including defined chain length and sulfation patterns of heparan sulfate/heparin and the related glycosaminoglycans keratan sulfate and dermatan sulfate (Supplementary Table 2). Strikingly, PCSK9 showed a remarkable selectivity for trisulfated heparan sulfate/heparin disaccharide repeats consisting of [4)-α-GlcN-6,N-disulfate(1 → 4)-α-IdoA-2-sulfate-(1→] as evident from the binding of the heparin oligos I and VII encompassing three and two repeats, respectively (Fig. 4a–b and Supplementary Table 2). A minimum of two repeats was required for efficient binding as no binding was observed for heparin oligo X consisting of only one repeat, and it is possible that the presence of additional repeats may further increase PCSK9 binding to such heparin oligos. The requirement of sulfate groups at the amino group of GlcN or of ester-linked sulfate groups at carbon 6 of GlcN and at carbon 2 of IdoA was also apparent by comparing the binding to heparin oligos I, II, and III, demonstrating that PCSK9 binding strictly requires repeating trisulfated disaccharides (Fig. 4a, b). Natural heparin with an average molecular weight of 5 kDa was included in the array as a positive control.

Remarkably, the binding signal for heparin oligo I was several folds higher than for natural heparin, which is a heterogeneous mixture of variable chain length and sulfation pattern.

**PCSK9 inhibitory antibodies directed at the HSPG-binding site.** We asked the question whether an antibody specifically directed against the HSPG-binding surface in PCSK9 could have a similar inhibitory effect as heparin and heparin mimetics. Thus, we generated PCSK9 antibodies by DNA immunization of rats using a construct encoding a rat chimeric protein encompassing the human HSPG-binding sequence stretch (Fig. 5a). Polyclonal antibodies that specifically recognized native human PCSK9 were successfully generated from three rats (Fig. 5b) and these showed a promising ability to increase LDLR in HepG2 cells (Fig. 5c). Monoclonal antibodies (mAbs) were subsequently generated, and a series of 12 inhibitory mAbs were obtained, which individually provided an up to 2-fold increase in LDLR levels in HepG2 cells (Fig. 5d, e). Of note, this magnitude is similar to the reported effect of mAbs directed against the LDLR-binding site[26, 27]. We next probed the 12 mAbs for their ability to immunoprecipitate metabolically labeled PCSK9 mut 5 (Fig. 5f). Clones 1G8, 5E11, and 8H4 showed no recognition of mut 5, showing that at least one the six basic acids depicted in Fig. 1b is part of the epitope of these PCSK9 inhibitory antibodies (Fig. 5f). Further analysis by immunoprecipitation (IP) of PCSK9 HSPG mutants showed that

clones 1G8 and 5E11 critically depended on R96, R97, R104, and R105 (Supplementary Fig. 7b). Finally, we found that mAb 5E11 increased the uptake of LDL in HepG2 cells to a similar extent as evolocumab directed at the LDLR-binding surface (Fig. 5g). Importantly, mAb 5E11 did not perturb the direct binding

between PCSK9 and LDLR in a cell-free assay (Supplementary Fig. 7c).

**PCSK9 activity in vivo requires HSPG binding.** To study the role of HSPG in PCSK9 activity in vivo, we tested the ability of

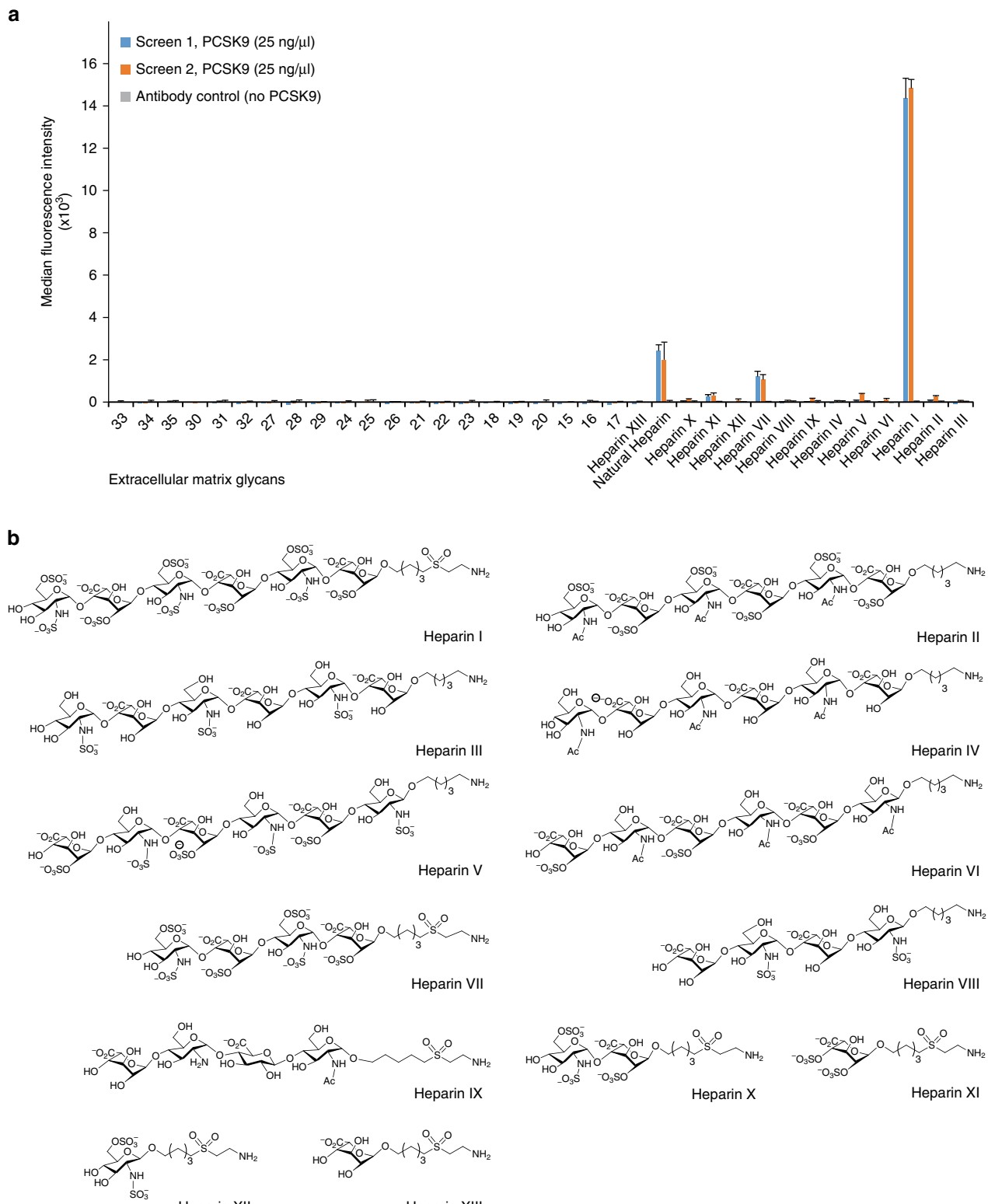

**Fig. 4** PCSK9 binding to synthetic glycans. **a** Dissection of the interaction between PCSK9 and extracellular matrix glycans (numbers refer to Supplementary Table 2) using a synthetic glycan microarray. Natural heparin is included as a positive control. **b** The heparin structures immobilized in the glycan microarray are shown. Structures interacting with PCSK9 contain repeats of [4)-α-GlcN-6,N-disulfate(1 → 4)-α-IdoA-2-sulfate-(1→]

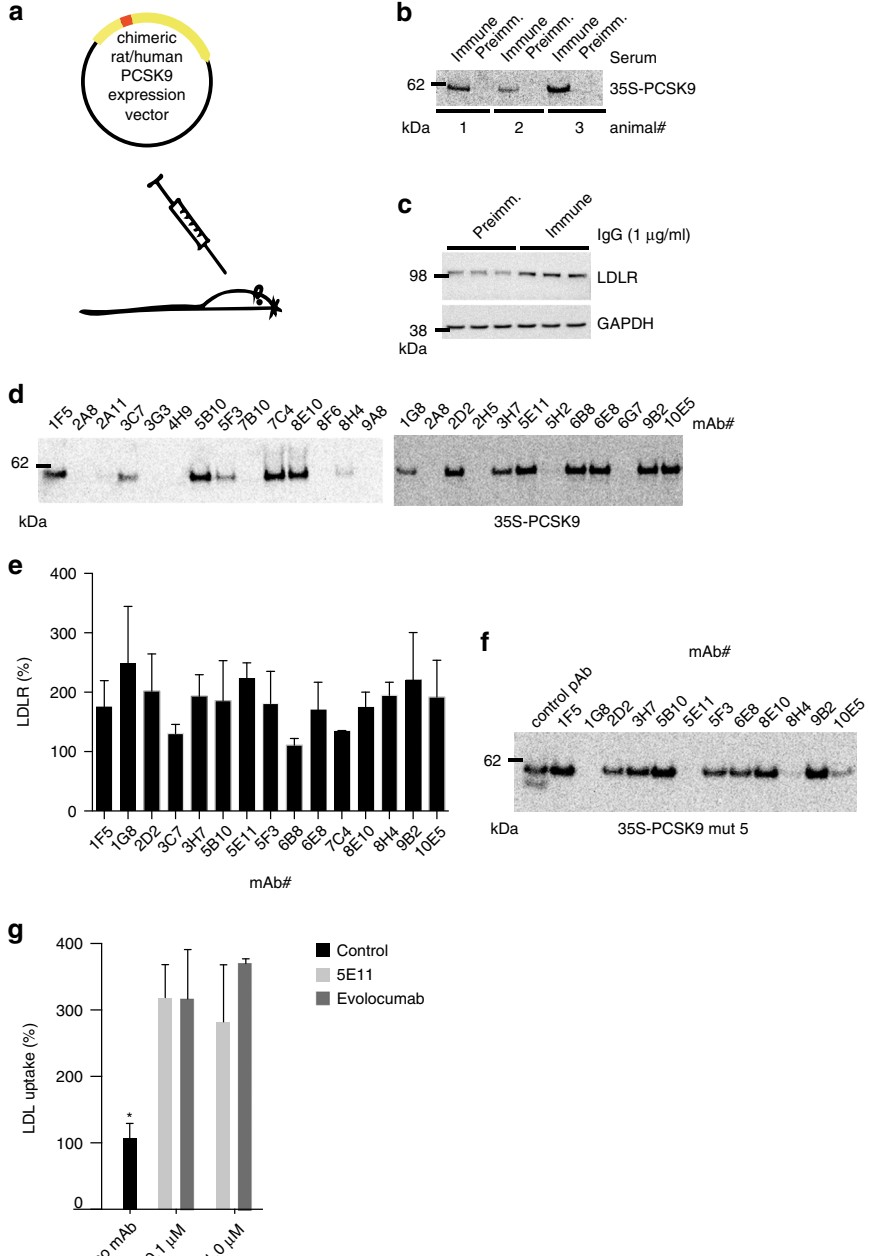

**Fig. 5** Generation of inhibitory mAbs directed against the PCSK HSPG-binding site. **a** Rats were immunized with cDNA encoding a rat PCKS9 chimera encompassing the human HSPG-binding site. The overall sequence identity between rat and human PCSK9 is 77%. **b** IP of radioactive-labeled PCSK9 (35S-PCSK9) using serum from three immunized rats. Preimmune serum is used as control. See full gel in Supplementary Fig. 7a. **c** LDLR levels in HepG2 cells following incubation with IgG from immunized animals compared to preimmune IgG (1 μg/ml). **d** mAbs were tested by IP. Fifteen mAbs specifically precipitated radioactive-labeled PCSK9. **e** LDLR levels in HepG2 cells incubated with the individual mAbs (*n* = 3–5). Twelve individual mAbs showed a significant and around 2-fold increase in LDLR, and were tested for binding to PCSK9 mut 5 by IP. **f** A polyclonal antibody was used as positive control (R&D systems, AF3888). Three mAbs, including 5E11, failed to precipitate PCSK9 mut 5. **g** Uptake of fluorescently labeled LDL (DiI-LDL) in HepG2 cells incubated with 100 nM PCSK9 alone (control) or in combination with 5E11 or evolocumab as indicated (*n* = 3). *Bar graphs* show mean with s.e.m. error bars. Statistical significance was evaluated using a two-tailed Student's *t*-test. Supplementary Fig. 10 shows uncropped gel images

tail vein-injected PCSK9 to downregulate LDLR with or without prior injection of heparinase I. PCSK9 alone (10 μg, corresponding to an initial concentration of approximately 68 nM) resulted in approximately 90% decrease in LDLR levels after 60 min specifically in the liver but not in the adrenal gland (Supplementary Fig. 8a, b). In contrast, injection of heparinase I 5 min before PCSK9 administration completely protected LDLR from degradation, demonstrating that HSPG are instrumental in PCSK9-induced LDLR degradation in vivo (Fig. 6a, b).

Heparinase treatment unmasks the antigenicity of syndecan-1, a major liver HSPG (Supplementary Fig. 9), confirming the effectiveness of the treatment (Fig. 6a). We subsequently tested if heparinase releases and antagonizes endogenous PCSK9 and found that heparinase injection resulted in approximately 30% increase in plasma PCSK9 15 min later (Fig. 6c, d). Interestingly, we observed a similar increase in plasma PCSK9 accompanied by markedly increased liver LDLR and reduced total plasma cholesterol in transgenic mice with constitutive overexpression of

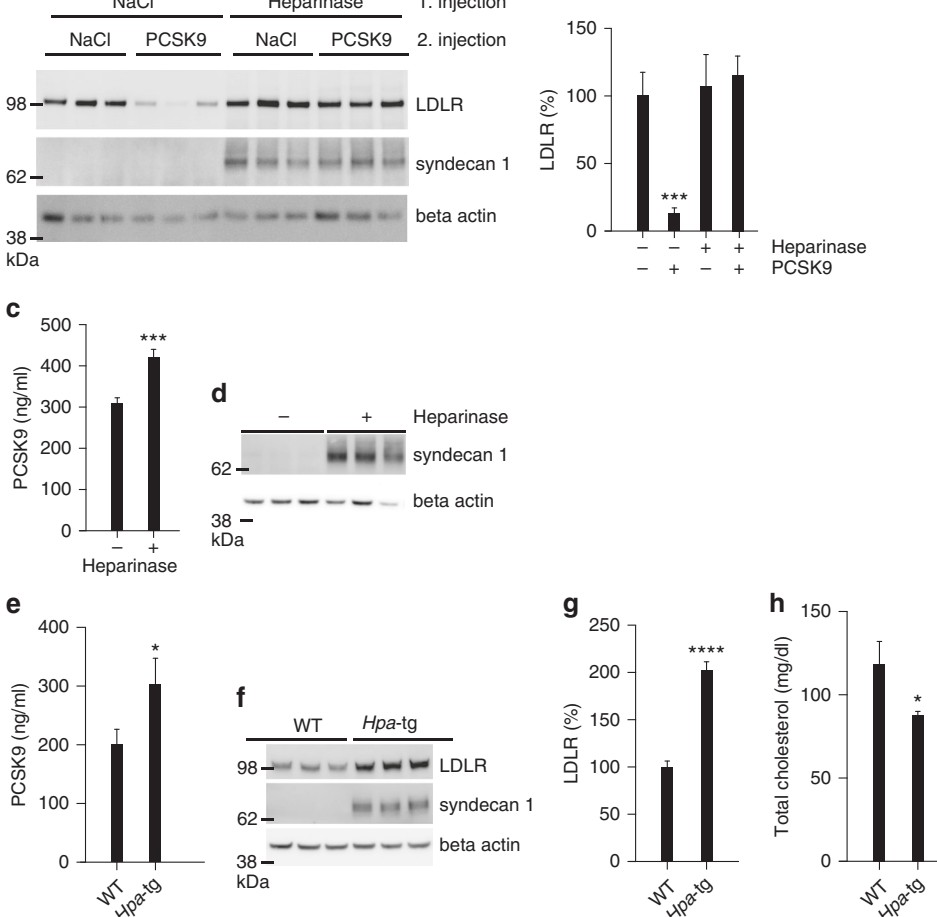

**Fig. 6** Enzymatic removal of liver heparan sulfate releases PCSK9 and ablates its activity. **a**, **b** Infusion of heparinase I prior to the injection of PCSK9 (10 µg) completely inhibits PCSK9-induced degradation of LDLR. Western blot of representative samples is shown in **a** and quantification of LDLR in **b** (control $n = 7$, PCSK9 $n = 6$, heparinase $n = 5$, heparinase/PCSK9 $n = 5$). Heparinase treatment unmasks the antigenicity of the major liver HSPG syndecan-1 (*middle panel*). Beta-actin is used as loading control (*lower panel*). **c** Heparinase I treatment leads to an increase in plasma PCSK9 as measured by ELISA 15 min after injection (control $n = 6$, heparinase $n = 6$). **d** Western blot (representative samples) of liver syndecan-1 is used as control of heparinase injection. Beta-actin is shown as loading control. **e–h** Transgenic mice with constitutive expression of human heparanase (*Hpa*-tg) ($n = 7$) have increased plasma PCSK9 (**e**), increased liver LDLR (**f**, **g**), and reduced plasma cholesterol (**h**) compared to control WT mice ($n = 6$). *Bar graphs* show mean with s.e.m. error bars. Statistical significance was evaluated using a two-tailed Student's *t*-test. Supplementary Fig. 11 shows uncropped gel images

human heparanase (*Hpa*-tg)[28], suggesting that HSPG binding is critical for endogenous PCSK9 activity (Fig. 6e–h).

We further found that both heparin and suramin prevented LDLR degradation as co-injection of heparin (500 µg) or suramin (300 µg) with PCSK9 had a pronounced protective effect on liver LDLR (Fig. 7a). In addition, a single injection of suramin alone (50 mg/kg) significantly increased LDLR levels and reduced total plasma cholesterol 18 h later specifically in WT mice but not in PCSK9 knockout mice (Fig. 7b, c), suggesting that the effect is indeed PCSK9-dependent. Protection of LDLR was also observed upon co-injection of PCSK9 with mAb 5E11 (50 µg), showing that directly targeting the HSPG site has inhibitory effect in vivo (Fig. 7d, e). Furthermore, PCSK9 with mutated HSPG site (mut 5) was unable to induce LDLR degradation (Fig. 7f, g). Similarly, while injection of PCSK9 WT significantly increased total cholesterol in mice on Western type diet, no such effect was observed by PCSK9 mut 5 (Fig. 7h), confirming that the HSPG binding is active in vivo. Previous studies have proposed that the clearance pathway for plasma PCSK9 is through LDLR[14, 29]. We observed that plasma levels of injected PCSK9 mut 5 were markedly increased compared to PCSK9 WT at the time points

analyzed (Fig. 7i) in line with reduced capture by HSPG and reduced presentation to LDLR. Hence, these results indicate that HSPG and LDLR may in fact cooperate in PCSK9 plasma clearance.

## Discussion

Our combined results lead us to propose a model (Fig. 8) in which HSPG on the hepatocyte surface capture PCSK9 and present it to the LDLR, hereby ensuring optimal conditions for PCSK9:LDLR complex formation. The transmembrane HSPG of the syndecan type undergo ligand-induced clustering in the plasma membrane[30–32] and mediate both clathrin-dependent and independent endocytosis of bound ligands[30, 31, 33–35]. Hence, it is tempting to speculate that sorting motifs in the cytoplasmic tail of syndecans could participate in directing the PCSK9:LDLR complex to lysosomes for degradation.

The HSPG-binding site is located in the PCSK9 prodomain in a region not previously implicated mechanistically in LDLR degradation and found on the opposite surface of the structure compared to the LDLR-binding site, which is located in the

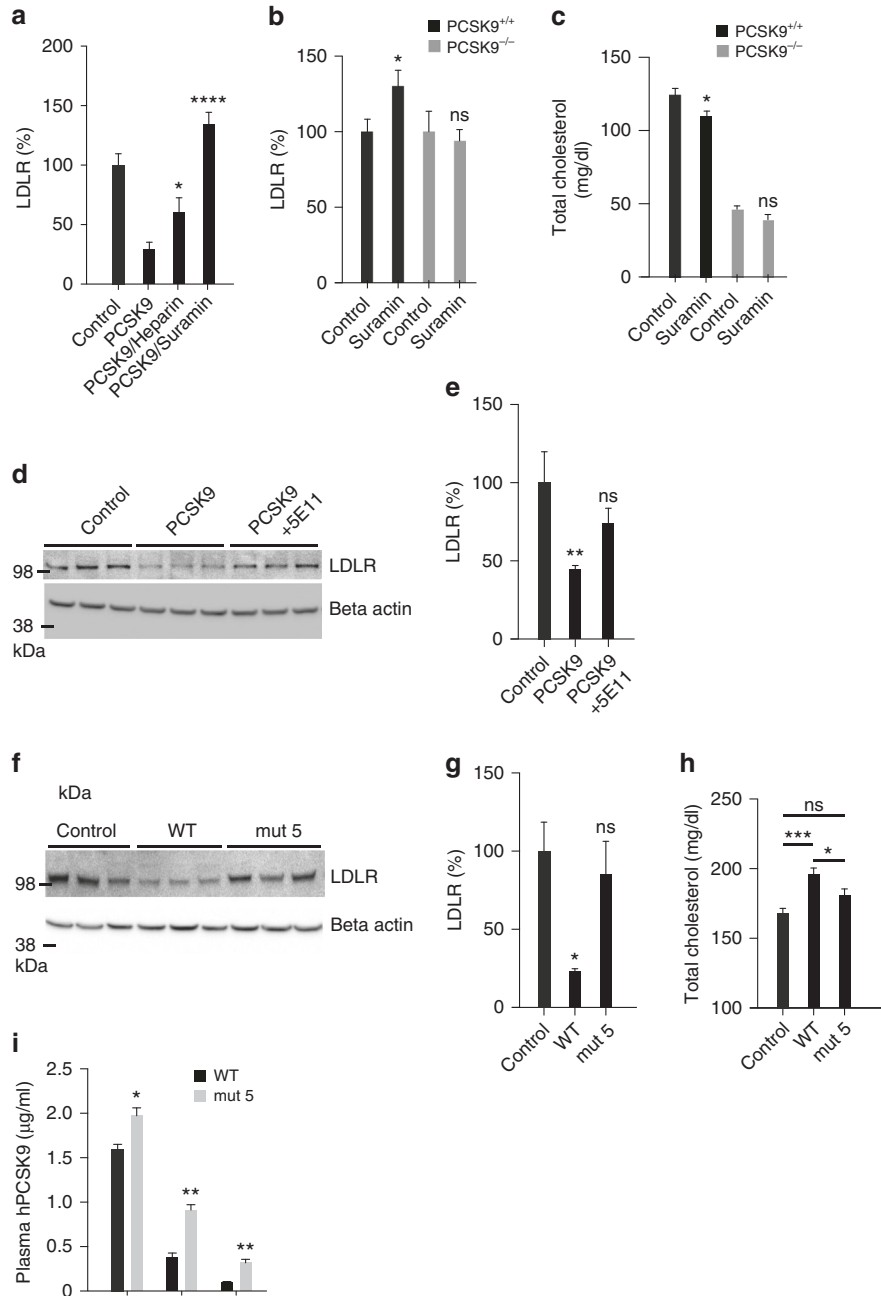

**Fig. 7** Inhibition of PCSK9:HSPG interaction protects LDLR in vivo. **a** LDLR is protected in mice co-injected with PCSK9 (10 μg) and heparin (500 μg) or suramin (300 μg) compared to mice injected with PCSK9 alone (control $n = 15$, PCSK9 $n = 10$, PCSK9/heparin $n = 7$, PCSK9/suramin $n = 3$). **b**, **c** A single injection of suramin (50 mg/kg) increased liver LDLR levels (**b**) and reduced total plasma cholesterol (**c**) 18 h later in WT mice (control $n = 5$, suramin $n = 8$) but not in PCSK9 KO mice (control $n = 5$, suramin $n = 7$). **d**, **e** Co-injection of PCSK9 (10 μg) with mAb 5E11 (50 μg) protects the LDLR from degradation (control $n = 4$, PCSK9 $n = 7$, PCSK9/mAb 5E11 $n = 6$). Representative blot is shown. **f**, **g** PCSK9 with mutated HSPG-binding site (mut 5) (10 μg) is ineffective in inducing LDLR degradation ($n = 3$ of each). **h** PCSK9 but not PCSK9 mut 5 significantly increases total cholesterol in mice fed Western-type diet 6 h post injection (control $n = 25$, PCSK9 $n = 12$, PCSK9 mut 5 $n = 9$). **i** Injected PCSK9 mut 5 remained in circulation for a prolonged time compared to PCSK9 WT in line with reduced capture by HSPG and clearance by LDLR. *Bar graphs* show mean with s.e.m. error bars. Statistical significance was evaluated using a two-tailed Student's $t$-test. Supplementary Fig. 11 shows uncropped gel images

inactive catalytic domain of PCSK9. HSPG binding involves six basic residues one of which is R93 (Fig. 1a, b and Supplementary Fig. 3d). Interestingly, individuals harboring a PCSK9 LOF variant in which R93 is substituted for cysteine (R93C) display highly significant reductions in LDL cholesterol and are protected against CAD[21, 22]. The molecular mechanism behind this PCSK9 LOF variant has so far been unknown, but loss of HSPG binding

as suggested by Fig. 1e may explain the low-LDL phenotype of R93C carriers.

The expression profile of HSPG encoding genes in the liver and the chemical properties of liver heparan sulfate GAG chains are unique[16], potentially providing the basis for the tissue-specific activity of PCSK9 as observed in Supplementary Fig. 8. As such, heparan sulfate chains isolated from rat liver are on average twice

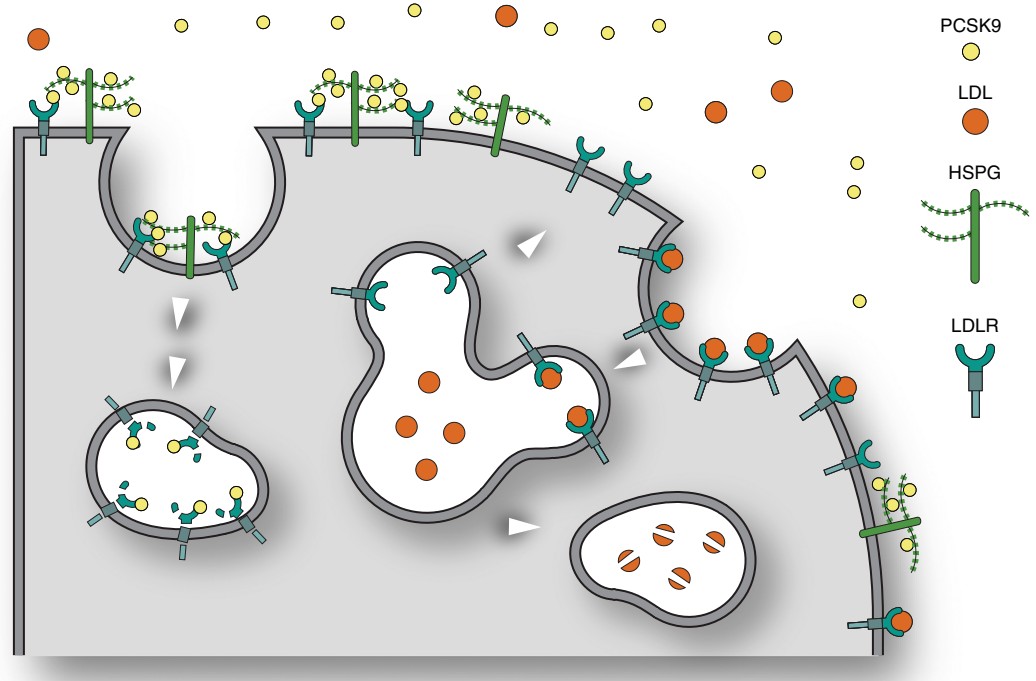

**Fig. 8** Model depicting the role of HSPG in PCSK9 activity in the hepatocyte. The model shows a hepatocyte and how LDLR in the absence of PCSK9 mediates endocytosis of bound LDL cholesterol particles followed by lysosomal degradation of LDL, while LDLR recycles to the cell surface. HSPG mediates the capture of PCSK9 and its subsequent presentation to LDLR thereby directing LDLR itself to lysosomes for degradation. In the model, we have depicted HSPG as syndecan but based on the present data we cannot exclude that it may also be a glypican

as negatively charged compared to other tissues and are enriched in trisulfated disaccharides[36]; the optimal structure for PCSK9 binding (Fig. 4a, b). Interestingly, highly sulfated HSPG on the hepatocyte surface are also employed by Plasmodium sporozoites during malaria infection and by both dengue and hepatitis C virus. Similarly to PCSK9, the interaction with heparan sulfate is mediated by specific binding motifs formed by basic residues on parasite and virus surface proteins[37–39], suggesting that this mechanism for liver-specific targeting is conserved in biology. As example, malaria infection is initiated by an Anopheles mosquito bite at the skin, injecting Plasmodium sporozoites that subsequently migrate on low-sulfated HSPG such as those found in skin and on the endothelial surface until highly sulfated HSPG on hepatocytes activates sporozoite invasion[37].

Liver heparan sulfate is covalently attached to two major classes of core proteins, syndecans (1–4) and glypicans (1–6). Isolated mouse hepatocytes express syndecan-1, syndecan-2, and syndecan-4 and glypican-1 and glypican-4[40]. Similar expression profile of syndecans was also observed in whole human and mouse liver by analyzing data from the two transcriptional databases Unigene (www.ncbi.nlm.nih.gov) and GTEx (www.gtexportal.org) (Supplementary Fig. 9). However, only glypican-3 is expressed in high levels in human but not in mouse liver according to Unigene. This expression was not confirmed by the GTEx data set and none of the other glypicans appeared to be expressed to any notable extent. Sulfation of heparan sulfate attached to one of these core proteins occurs in patterns generated by the cooperation of several different sulfotransferases. In the case of four key sulfotransferases Ndst (1–4)[41], a previous study using semi-quantitative PCR reported Ndst-1, Ndst-2, and Ndst-3 expression in human liver[42]. Ndst-1 and Ndst-2 but not Ndst-3 expression in human liver was confirmed by the Unigene

and GTEx data sets, while mouse liver was found to express Ndst-1 and Ndst-3 but not Ndst-2 (Supplementary Fig. 9). Future studies should address the specificity of the interaction between PCSK9 and the individual syndecans and glypicans, how these contribute to PCSK9-induced sorting of LDLR to lysosomes, and further which sulfotransferases are critical for generating the PCSK9 binding pattern in heparan sulfate chains in mouse and human cells.

Considering our findings, it is highly interesting that five clinical trials using long-term treatment with low-dose heparin at sub therapeutic amounts with respect to anticoagulation, all have reported marked reductions in cardiac events and deaths although the mechanism for this has remained unclear[43–48]. Furthermore, other studies have found that both acute and long-term heparin and dextran sulfate administration in humans reduces LDL and total cholesterol levels, respectively[49–54]. At a stage where patients throughout the world are about to be treated with humanized antibodies directed against the PCSK9:LDLR complex, the present study provides important insight into the biological mechanism of PCSK9 activity. It also opens up a venue for future development of small molecule heparin mimetics as PCSK9 inhibitors.

## Methods

**Cell culture.** Human hepatocyte-derived cells (HepG2) and human embryonic kidney cells (HEK293) were cultured at 37 °C in a humidified atmosphere containing 5% $CO_2$ in Dulbecco's Modified Eagle's Medium (DMEM), supplemented with 10% fetal calf serum, penicillin (50 U/ml), and streptomycin (50 µg/ml). CHO-K1 cells were cultured in HyClone-CCM5 media (Thermo) supplemented with penicillin (50 U/ml) and streptomycin (50 µg/ml). Cells were transfected using Fugene6 transfection reagent (Promega). CHO-K1 cells stably transfected with PCSK9 cDNA were maintained in 400 µg/ml G418, and stably transfected HepG2 cells in 1 µg/ml puromycin (InvivoGen). All cells were obtained from ATCC and routinely checked for mycoplasma infection.

**DNA constructs and recombinant protein expression**. The peptide sequence of PCSK9 used in this study is identical to GenBank acc. no. CAC38896.1. The PCSK9 coding sequence was ligated into the pCpGfree-vitroNmcs expression vector and transformed into the *E. coli* strain GT115 encoding the *pir* gene (Invivogen). Mutants were created by successive overlapping PCR and inserted into the pCpG-vitroNmcs vector. Plasmids were sequenced by Eurofins. WT PCSK9 and derived mutants were expressed transiently in CHO-K1 cells. CHO-K1 cells stably trans-fected with PCSK9 were adapted to growth in suspension in Hybridoma-SFM medium (GIBCO) and then expanded in Celline CL 350 Bioreactor flasks (Integra) or in triple layer flasks (Thermo).

**Purification of recombinant PCSK9**. Growth media from PCSK9-expressing CHO cells were dialyzed overnight against 20 mM Tris-HCl pH 7.6 and 75 mM KCl. The dialyzed sample was loaded on a 1 ml MonoQ column (GE Healthcare) equilibrated in buffer Q-A (20 mM Hepes pH 7.6, 75 mM KCl, and 10% glycerol). Bound PCSK9 was eluted with a gradient from 0 to 100% buffer Q-B (20 mM Hepes pH 7.6, 500 mM KCl, and 10% glycerol). Fractions containing PCSK9 were pooled and further purified on a 24 ml Superdex 200 INCREASE column (GE Healthcare) equilibrated in phosphate buffered saline (PBS). Fractions containing PCSK9 were pooled and concentrated to 1 mg/ml (based on absorbance at 280 nm) using a centrifugation filter (Amicon Ultra-4 10,000 MW cut-off). The con-centration of PCSK9 was subsequently determined by ELISA (R&D Systems, DPC900).

**Surface plasmon resonance analysis**. Surface plasmon resonance analysis was performed on a Biacore 3000 equipped with CM5 sensor chips activated. LDLR extracellular domain (Biaffin GmbH & Co KG, LDR-H5224) was immobilized to a density of 60 fmol/mm$^2$ in 10 mM sodium acetate, pH 4.0, and remaining coupling sites were blocked with 1 M ethanolamine, pH 8.5. Samples, PCSK9 WT and mut 5, were injected at 5 μl/min at 25 °C in 10 mM HEPES, pH 7.4, 150 mM NaCl, 1.5 mM CaCl$_2$, 1 mM EGTA, and 0.005% Tween 20 (CaHBS). Binding was expressed in relative response units; the difference in response between the immobilized protein flow cell and the corresponding control flow cell. Kinetic parameters were determined using BIAevaluation 4.1.

**Heparin affinity chromatography**. PCSK9 was loaded onto a 5 ml HiTrap Heparin HP column (GE Healthcare) in PBS. The column was connected to a Äkta Prime and washed with 5 column volumes of 10 mM NaH$_2$PO$_4$ (pH 7.4). PCSK9 was eluted using a linear gradient of 10 mM NaH$_2$PO$_4$ (pH 7.4) and 2 M NaCl and fractions were analyzed by SDS-PAGE. Based on the measured con-ductivity, the elution profile was transformed to a function of NaCl concentration using the transformation coefficient 0.065 mS/mM NaCl. In a different experiment, conditioned media from HepG2 cells or CHO cells transfected with PCSK9 var-iants of interest were incubated with Heparin Sepharose CL-6B beads (GE Healthcare) in 10 mM NaH$_2$PO$_4$ (binding buffer). Following overnight incubation on a rotor at 4 °C, beads were washed in binding buffer before batch elution of heparin-bound proteins in increasing concentration of NaCl. PCSK9 in the input, flow through, and elution fractions was determined by Western blotting.

**Antibodies**. The following antibodies were used for Western blotting: anti-LDLR (Abcam ab52818, 1:1000), anti-GAPDH (Sigma Aldrich G8795, 1:2000), anti-beta actin (Sigma A5441, 1:5000), anti-PCSK9 (R&D Systems AF3985, 1:1000), anti-syndecan-1 (R&D AF3190, 1:1000), and immunostainings: anti-PCSK9 (R&D Systems AF3888, 2 μg/ml), and fluorescent secondary antibodies (Invitrogen). Detailed information on our use of primary antibodies can be found in the pAbm-mAbs database www.pAbmAbs.com deposited by Camilla Gustafsen or Ditte Olsen.

**Production of PCSK9 inhibitory antibodies**. Antibodies raised against the HSPG-binding domain in PCSK9 were produced by Aldevron Freiburg, Germany. Rats were immunized with DNA encoding chimeric rat PCSK9 containing the human HSPG-binding domain. Serum samples were tested for PCSK9 binding by IP and their ability to increase LDLR levels in HepG2 cells. mAb-producing hybridoma clones were obtained by fusion of spleen lymphocytes with myeloma cells and subsequent single-cell subcloning of antibody-producing cells.

**Western blotting**. Proteins were separated by SDS-PAGE (NuPAGE 4–12% Bis-Tris; Invitrogen) and transferred to nitrocellulose membranes using iBlot Gel Transfer Stacks (Invitrogen). Membranes were blocked in 5% skimmed milk in TBST (0.05 M Tris-base, 0.5 M NaCl supplemented with 0.1% Tween-20), and incubated with primary antibodies diluted in blocking buffer overnight. Membranes were washed three times in 0.5% skimmed milk in TBST and incu-bated with horseradish peroxidase (HRP)-conjugated secondary antibodies. Fol-lowing a final washing step, proteins were visualized with the ECL plus Western blotting detection system (GE Healthcare) using a Fuji film LAS4000 system. See Supplementary Figs. 10 and 11 for uncropped Western blot images.

**Microscale thermophoresis**. Equilibrium-binding affinities between PCSK9 and ligands (suramin, dextran sulfate 5000, dextran 5000, pentosan sulfate, and S-dC-36) were assessed using MST[55]. PCSK9 was labeled using the MO-L003 Monolith Blue-NHS labeling kit (NanoTemper Technologies) and a labeling efficiency of 1:1 molar ratio of protein to dye was achieved. PCSK9 was applied at a final con-centration of 100 nM. The unlabeled binding partner was titrated in 1:1 dilutions (in PBS + 0.05% Tween-20) where the highest concentrations were 15.4 mM for suramin, 2.3 mM for dextran sulfate 5000, 2.3 mM for dextran 5000, 16.3 mM for pentosan sulfate, and 250 μM for S-dC-36. MST measurements were performed in standard-treated capillaries (NanoTemper Technologies) on a Monolith NT.115 instrument (NanoTemper Technologies) using 20% LED and 80% MST power. Laser on and off times were 5 and 35 s, respectively. Negative controls were per-formed using 100 nM of labeled PCSK9 in MST buffer in all 16 capillaries under the same conditions as mentioned above. Binding curves were obtained from the temperature-jump phase at 80% MST power for suramin, dextran sulfate 5000, and S-dC-36; and from the thermophoresis + temperature-jump phase at 80% MST power for pentosan sulfate. For each binding partner, the sigmoidal dose-response curves were fitted with GraphPad Prism 6 to yield an average $K_D$ value. As fluorescence quenching of PCSK9 was observed in the presence of high con-centrations of suramin, a denaturation test was performed by pre-treating PCSK9 with 10% SDS and then heating the samples for 15 min at 90 °C prior to analysis on the MST apparatus. This eliminated the quenching effect, indicating that the fluorescence quenching of PCSK9 under non-denaturing conditions was due to an actual ligand-binding event.

**PCSK9-induced LDLR degradation in vitro**. HepG2 cells have several morpho-logical and functional features of human hepatocytes, including endogenous expression of LDLR and PCSK9. We continuously confirmed this prior to per-forming experiments. HepG2 cells were seeded at a density 250,000 cells per well in 12-well plates. After overnight incubation, the media was replaced with fresh media containing one of the following components: purified PCSK9 variants, PCSK9 mAbs, heparin (Leo Pharma, 585679), glycosaminoglycans: heparan sulfate (Sigma, H7640) or chondroitin sulfate (Sigma, C9819), LMWH: Fragmin (Pfizer, 085423) and Innohep (Leo Pharma, 595570), heparin mimetics: suramin sodium (Sigma, S2671), dextran sulfate sodium (Sigma, D4911), pentosan sodium polysulfate (Anthropharm Ireland limited, 003192), and phosphorothioate oligodeoxycytidine 36-mer (S-dC-26) (Sigma). Cells were harvested and lysed following 18 or 24 h incubation and LDLR levels were assessed by Western blotting and quantification by densitometry.

**LDL uptake assay**. HepG2 cells were seeded at 25,000 cells/96 well. Following overnight incubation, the medium was changed to DMEM without supplements for an additional 24 h. Cells were subsequently incubated with or without PCSK9 (100 nM) and with or without PCSK9 inhibitors 5E11 (0.1–1.0 μM), evolocumab (0.1–1.0 μM) and suramin (200 μg/ml) in serum-free medium for 4 h at 37 °C before addition of 5 μg/ml DiI-LDL (Thermo Fisher Scientific) or BODIPY-LDL (Thermo Fisher Scientific) for 4 h at 37 °C. Wells were washed with PBS and cellular uptake of fluorescence was evaluated using an EnSpire Alpha Plate Reader (Perkin Elmer) at excitation/emission 552/573 nm (DiI) or 515/520 nm (BODIPY). After overnight incubation of plates at −80 °C, the number of cells/well was quantified using the CyQuant Cell Proliferation Assay (Thermo Fisher Scientific), according to the manufacturer's protocol.

**Mouse experiments**. Unless otherwise stated, in vivo experiments were performed with 10–12-week-old male BALB6/cJRj (Janvier Labs). Mice were injected intra-venously (tail vein) with 0.9% saline (vehicle control), PCSK9 (10 μg) with or without heparin (50 units), suramin (300 μg), or mAb 5E11 (50 μg) in a rando-mized order by an investigator blinded to the content. Heparinase I (30 U) (Sigma Aldrich/H2519) was administrated though a tail vein catheter 5 min before injec-tion of PCSK9 (10 μg). During heparinase infusion, mice were lightly anaesthetized with isoflurane continuously administered through a mask. One hour post injec-tion, mice were sacrificed and tissue samples were collected and snap frozen. Proteins were extracted from liver and adrenal gland and the concentration was determined (Bicinchoninic Acid Assay (Sigma)). Equal amounts of samples (30 μg protein) were analyzed by Western blotting. In a different experiment, suramin (50 mg/kg) was injected in BALB6/cJRj and PCSK9-deficient control mice (B6;129S6-Pcsk9$^{tm1Jdn}$/J, Jackson Laboratory)[56], and tissue samples were collected 18 h after injection. *Hpa*-tg mice overexpressing human heparanase driven by a β-actin promoter in a BALB/c genetic background were generated by breeding with WT Balb/C from Harlan Laboratories[28]. Plasma and liver tissue samples were obtained from 8-weeks male animals. All animal experiments were carried out according to the institutional ethical guidelines of Aarhus University and approved by the Danish Animal Experiments Inspectorate under the Ministry of Justice.

**Plasma cholesterol measurement**. Male BALB6/cJRj mice (Janvier Labs) were fed ad libitum western-type diet (D12079B, Research Diet) for 4 weeks before injection of WT or mutant PCSK9 (mut 5) (10 μg/25 g mouse). Total cholesterol levels in Li-Heparin plasma sampled prior (control) and six hours post injection were assessed using Cholesterol CHOD-PAP reagent (Roche/Hitachi). Plasma cholesterol levels

in suramin-injected BALB6/cJRj and PCSK9 KO mice, as well as in *Hpa*-tg mice, were determined in a similar manner.

**Immunoprecipitation.** Antibodies from 1 μl immune or preimmune rat serum or 500 μl conditioned media from mAb-producing hybridoma clones were immobilized on GammaBind beads (GE Healthcare), and used for precipitation (3 h at 4 °C) of $^{35}$S human PCSK9 from conditioned media of metabolically labeled HEK293 transiently transfected with PCSK9 as described previously[9]. Following three washes in TBS containing 0.1% Triton X100, precipitated proteins were eluted by boiling samples in NUPAGE sample buffer supplemented with 20 mM dithioerythritol, separated by SDS-PAGE, and visualized by phosphorimaging.

**Crystal structure of PCSK9:dextran sulfate.** PCSK9 was previously co-crystalized with a Fab fragment of evolocumab in a crystallization cocktail composed of 0.1 M Tris (pH 8.3), 0.2 M sodium acetate, 10–15% PEG4000, and 3–6% dextran sulfate sodium salt (Mr: 5000)[26]. The coordinates and corresponding structure factors from the Protein Databank (PDBid: 3H42) was used to examine the structure. A difference electron density (Fo-Fc) map revealed a peak 11 $\sigma$ above the mean density near the alkaline helix (residues 92–104) in the pro-domain of PCSK9 which was not accounted for in the deposited model. Next a simulated annealing composite omit map was created to remove model bias. The omit map showed additional unaccounted electron density near the alkaline helix (Fig. 3c and Supplementary Fig. 6). A disaccharide of dextran sulfate was placed into this electron density and the structure was refined by iterative turns of model building in *Coot*[57], followed by subsequent refinement using phenix.refine[58]. To confirm that the improved agreement between model and data was due to the inclusion of dextran sulfate, we added the refined position of dextran sulfate only to the deposited model and calculated its refinement statistics (Supplementary Table 1). The dextran sulfate disaccharide was placed in two alternative orientations in the electron density. The occupancy and real-space correlation coefficient of the two conformations were 40/74% and 30/73% for conformation 1 and conformation 2, respectively. The overall B-factor for conformations 1 and 2 was 78 Å$^3$. For comparison, the surrounding residues showed similar B-factors, i.e., 79 Å$^3$ for R104, 61 Å$^3$ for R97, 62 Å$^3$ for R93, and 64 Å$^3$ for H139. For simplification, conformation1 with the highest occupancy is shown in Fig. 3a, b.

**Extracellular matrix glycan microarray.** Extracellular matrix glycan microarray, including synthetic oligosaccharides of heparan sulfate/heparin, keratan sulfate, dermatan sulfate, and 5 kDa natural heparin (Santa Cruz) (Supplementary Table 2), were prepared as described previously using 250 μM spotting solution[59, 60]. After spotting, the microarray slides were incubated overnight in a humid chamber, followed by three washing steps with water and quenching with 50 mM ethanolamine solution, pH 9, for 1 h at 50 °C to remove the remaining reactive groups. Slides were washed three times with water, blocked at room temperature for 1 h with 1% (w/v) BSA in PBS, pH 7.4 and dried by centrifugation (5 min, 300 × *g*). PCSK9 (25 μg/ml) were incubated overnight at 4 °C diluted in 1% (w/v) BSA-PBS in a humid chamber. After three washing steps with 0.1% (v/v) Tween-PBS, an incubation with preincubated 0.1 μg/ml humanized anti-PCSK9 mAb and 5 μg/ml goat anti-human IgG Alexa Fluor 647 (Life Technologies) diluted in 1% BSA-PBS was performed for 1 h at room temperature. Slides were washed three times with 0.1% (v/v) Tween-PBS, rinsed once with water and dried by centrifugation. For data acquisition slides were scanned with GenePix 4300 microarray scanner (Molecular Devices, Sunnyvale, CA, USA) by exciting Alexa Fluor 647 at 635 nm. Photomultiplier tube values were set to reveal scans free of saturation. Median fluorescence values were determined with GenePix Pro 7 software (Molecular Devices).

**Immunofluorescence.** HepG2 cells stably transfected with PCSK9 were seeded at 50,000 cells per coverslip and incubated overnight before addition of Heparinase I (Sigma Aldrich/H2519) or Chondroitinase (Sigma Aldrich/C3667) in PBS (0.0002 U/ml) or PBS alone. Cells were incubated for 1 h at 37 °C, before fixation in 4% paraformaldehyde and immunostaining of non-permeabilized cells with primary and secondary antibodies. Nuclei were visualized with Hoechst dye (Sigma Aldrich). Images were acquired on a Zeiss LSM780. In a different experiment, HepG2 cells were incubated with Heparinase I before washing with 3× PBS and incubation with PCSK9 (500 nM) on ice. Following binding of PCSK9 cells were fixed and non-permeabilized cells were subjected to immunofluorescence staining.

**Proximity ligation assay.** The PLA (Duolink®II, Olink Bioscience) was performed according to manufacturer's protocol using anti-PCSK9 (R&D Systems/AF3888), and anti-LDLR (Abcam/ab52818) as primary antibodies. PCSK9 and LDLR located within 30 nm from each other are visualized by oligonucleotide-conjugated secondary antibodies that hybridize with circle-forming oligonucleotides thereby priming rolling circle amplification. The amplified DNA is visualized by addition of complementary fluorescently labeled oligonucleotides.

**Molecular docking of SANORG.** Coordinates for the heparin pentasaccharide (SANORG) were extracted from the crystal structure of α-antithromin III (PDB ID

1E03)[61]. PCSK9 (PDB ID 3H42)[26] and heparin were prepared for docking, i.e., adding Gasteiger charges and hydrogens, with *AutoDocktools*[62]. Docking was performed with *Autodock Vina*[63] using a grid with dimensions of 43 × 46 × 40 centered at R97. During docking, residues R93, R96, R97, R104, R105, and H139 were considered flexible. Ligand poses were scored according to their calculated free energy and the pose with lowest free energy while also fulfilling the restraint to contain sulfate groups coordinated by H139, and R93, R97 and R104 was chosen. PyMol (Schrödinger, LLC) was used for modeling and visualizing structures.

**Enzyme-linked immunosorbent assay.** The PCSK9 concentration in conditioned media of HepG2 cells was measured using Human (DPC900) Quantikine ELISA kit from R&D Systems, according to the manufacturer's protocol. The ELISA kit was also used to assess the concentration of purified PCSK9 WT and mut 5 used for in vitro and in vivo experiments.

**Quantitative RT-PCR.** RNA extraction from HepG2 cells was performed using NucleoSpin RNA preparation kit (Macherey-Nagel), following cDNA synthesis from 0.5 μg RNA template by iScript™ cDNA synthesis kit (BIORAD). Real-time PCR was performed with iQ SYBR® Green supermix and iTaq™ polymerase using the following primers to detect transcripts of LDLR (forward primer 5′-ACGGC GTCTCTTCCTATGACA-3′, reverse primer 5′-CCCTTGGTATCCGCAAC AGA-3′), PCSK9 (forward primer 5′-CCTGGAGCGGATTACCCCT-3′, reverse primer 5′-CTGTATGCTGGTGTCTAGGAGA-3′), and GAPDH (forward primer 5′-ACAACTTTGGTATCGTGGAAGG-3′, reverse primer 5′-GCCATCACGCCA CAGTTTC-3′).

**PCSK9/LDLR-binding assay.** The interaction between PCSK9 and LDLR in the presence of heparin was analyzed using a PCSK9-(biotinylated)-LDLR-binding assay kit (BPS Bioscience), according to the manufacturer's protocol. Briefly, microtiter plate wells coated with LDLR extracellular domain were incubated with biotinylated PCSK9 in the presence of heparin (5 or 50 U/ml), 5 nManti-PCSK9 (BPS Bioscience #71207), evolocumab (Amgen), or mAb 5E11. Following washing, wells were incubated with HRP-labeled streptavidin and binding of PCSK9 to the LDLR extracellular domain was assessed by addition of HRP substrate and evaluation of signal using a chemiluminescence microplate reader.

**Statistics.** Results were evaluated using Student's *t*-test and error bars indicate standard error of mean (s.e.m.). All data were normal distributed and the variance was similar among groups. *p*-values are indicated by asterisks: *$p < 0.05$, **$p < 0.01$, ***$p < 0.001$, ****$p < 0.0001$, and *****$p < 0.00001$.

**Data availability.** The coordinates of the PCSK9:dextran sulfate structure have been deposited in the Protein Data Bank (PDB ID: 5OCA). Antibody testing data has been posted on www.pabmabs.com by Camilla Gustafsen or Ditte Olsen. The data sets generated in the current study are included in this published article or are available from the corresponding authors upon reasonable request.

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

## Acknowledgements

The present study is supported by the Danish Council of Independent Research Sapere Aude starting grant (S.G., grant number DFF 4183-00604), the Novo Nordisk Foundation (S.G.), the Danish Heart Association (C.G.), and Jens Anker Andersen Foundation (S.G.). P.H.S. thanks the Max-Planck Society, the ERC (Advanced Grant AUTO-HEPARIN), and the Deutsche Forschungsgemeinschaft (TRR67) for generous financial support. We thank Mette Singers, Mitra Shamshali, and Anne Marie Bundsgaard for excellent technical support.

## Author contributions

S.G. and C.G. provided funding for the study, designed experiments, and wrote the manuscript. C.G., D.O., J.V., S.L., A.R., N.W., T.L., C.B.F.A., P.M. conducted research. K. W., J.-P.L., P.H.S., S.T., P.M. designed experiments and contributed to the discussion.

## Additional information

**Competing interests:** C.G., P.M., and S.G. are inventors on two patent applications on compounds for treating lipoprotein metabolism disorders submitted by Aarhus University. C.G., P.M., and S.G. have significant financial interest in Draupnir Bio ApS, a company that develops PCSK9 inhibitors and has exclusively licensed the above-mentioned intellectual property. The remaining authors declare no competing financial interests.

