## [Peer Review file · Nature Communications]

Reviewers' comments:

Reviewer #1 (Remarks to the Author):

Manuscript authored by Gustafsen and colleagues describes the role of heparan sulfate in participating in LDL clearance. The LDL receptor binds to LDL, responsible for clearing LDL from the circulation. It is known that PCSK9 interacts with LDL receptor, causing the receptor internalization, and consequently reduces the clearance of LDL. Blocking the interaction of LDL receptor and PCSK9 can improve the clearance of LDL. In this manuscript, authors discover that the interaction of PCSK9 and LDL receptor also involves heparan sulfate. Several lines of evidence are provided to support this novel discovery.

1. Heparan sulfate binding site in PCSK9 were identified. Mutation the basic amino acid residues in the binding site led the loss of heparin binding as demonstrated using heparin affinity chromatography.

2. Authors showed that the binding of PCSK9 and HS is required for the LDL receptor internalization. The PCSK9 mutant that is defect in binding to HS was unable to induce the degradation of LDL receptor. Treatment of heparanase, an enzyme that degrades heparan sulfate from the cell surface, also diminished the LDL receptor degradation. However, the treatment of chondroitinase of cell had no effect, demonstrating the structural selectivity for PCSK9 binding.

3. Heparin and heparin mimetic inhibits the binding of PCSK5 and LDL receptor. Authors suggest that highly sulfated domain of HS, which is abundantly present in the liver, is required for the binding.

4. Removal of HS by injecting heparinase I to mice protects the degradation of LDL receptor induced by PCSK9, offering an in vivo proof.

This is an important piece work to understand the biology of the LDL clearance process. The results also provide a potential therapeutic method to lower the level of LDL. There will be a wide range of audience who is interested in learning this result. The experiment design is solid, and the conclusion is supported by the data. No weakness can be identified in the current manuscript.

Reviewer #2 (Remarks to the Author):

This manuscript addresses the provocative concept that HSPGs serve as a receptor for PCSK9 in the liver and facilitate presentation of PCSK9 to the LDLR. They present data consistent with the concept that interruption of the PCSK9/HSPG interaction results in reduced PCSK9 activity and therefore could be a novel strategy for PCSK9 inhibition. The studies are interesting and fairly compelling with regard to the HSPG-binding of PCSK9.

Specific comments:

1. Do the antibodies against the PCSK9 HSPG binding site affect direct binding of PCSK9 to the LDLR in absence of HSPGs?

2. Does injection of WT PCSK9 affect LDLR protein in non-hepatic tissues? Is this effect altered by co-injection of heparin or suramin?

3. The majority of the in vivo studies employed the injection of exogenous PCSK9. Do heparinase and heparin increase the hepatic LDLR protein due to inhibition of endogenous PCSK9?

4. The only experiment relying on endogenous PCSK9 is represented by Fig 6de, an experiment involving a single injection of suramin in WT and PCSK9 KO mice. The changes in LDLR and total cholesterol are very modest (and in WT mice total cholesterol primarily represents HDL cholesterol). More evidence that this pathway is relevant to endogenous PCSK9 in vivo would substantially strengthen the manuscript.

Reviewer #3 (Remarks to the Author):

NCOMMS1702733T

This manuscript describes the presentation of PCSK9 to the LDL-R by a heparan sulfate (HS) proteoglycan (PG). This is a well-conducted study and an impactful paper. The results presented in the six data figures are strongly supportive of the author's conclusions. Figure 1 clearly confirms the binding of heparin, although the x-axes on panels k,l,m should be revised to present the heparin concentrations in micrograms/ml instead of units/ml as units of anticoagulant activity are completely unrelated to the activities discussed in this manuscript. Also the text suggests that PCSK9 contains a "perfect" binding site—what is a perfect binding site? In Figure 2 it is unclear why S-dC-38 was tested some more explanation is required here. Figure 3 is and the reinterpretation of the x-ray crystal structure is very convincing. Figure 4 suggests to this reviewer that three trisulfated disaccharide repeating units (present in the heparin I structure—this reviewer does not think it appropriate to refer to these oligos as heparins, maybe heparin oligo I would be more appropriate) bind tightly to PCSK9 but would four (or five) repeats bind even more tightly? Also why is natural heparin a weaker binder? Some more discussion on these points would be helpful. In Figure 5 some information on the differences between rat and human PCSK9 would be useful. In Figure 6 why was suramin selected and not the heparin oligo I structure used? The implications of this study are profoundly important as the discussion section points out. Therefore, a better cartoon for Figure 7 is warranted. The current version of Figure 7 clearly shows capture but not presentation. Furthermore, the discussion of the liver HS-PG is lacking and requires a more thorough discussion. There is substantial literature on the HS-PG as a liver ApoE co-receptor as well as involved in the tropism of malaria (through binding the CS protein) and hepatitis C and dengue (through binding their envelop proteins). These interactions all involve multiple trisulfated disaccharide repeating units. Are these all using the same HS-PG? Is this HS-PG a glypican or a syndecan (your cartoon structure in Figure 7 implies an integral transmembrane protein—a syndecan)? Or is the liver simply biosynthesizing highly sulfated HS glycosaminoglycan chains on all its HS-PGs?

Response from authors:

We would like to acknowledge the reviewers for providing an insightful and fair revision of our manuscript. We find their comments highly encouraging and they have served to improve the study. Below, we have provided a response to the individual comments point by point.

Reviewer #1 (Remarks to the Author):

Manuscript authored by Gustafsen and colleagues describes the role of heparan sulfate in participating in LDL clearance. The LDL receptor binds to LDL, responsible for clearing LDL from the circulation. It is known that PCSK9 interacts with LDL receptor, causing the receptor internalization, and consequently reduces the clearance of LDL. Blocking the interaction of LDL receptor and PCSK9 can improve the clearance of LDL. In this manuscript, authors discover that the interaction of PCSK9 and LDL receptor also involves heparan sulfate. Several lines of evidence are provided to support this novel discovery.

1. Heparan sulfate binding site in PCSK9 were identified. Mutation the basic amino acid residues in the binding site led the loss of heparin binding as demonstrated using heparin affinity chromatography.
2. Authors showed that the binding of PCSK9 and HS is required for the LDL receptor internalization. The PCSK9 mutant that is defect in binding to HS was unable to induce the degradation of LDL receptor. Treatment of heparanase, an enzyme that degrades heparan sulfate from the cell surface, also diminished the LDL receptor degradation. However, the treatment of chondroitinase of cell had no effect, demonstrating the structural selectivity for PCSK9 binding.
3. Heparin and heparin mimetic inhibits the binding of PCSK5 and LDL receptor. Authors suggest that highly sulfated domain of HS, which is abundantly present in the liver, is required for the binding.
4. Removal of HS by injecting heparinase I to mice protects the degradation of LDL receptor induced by PCSK9, offering an in vivo proof.

This is an important piece work to understand the biology of the LDL clearance process. The results also provide a potential therapeutic method to lower the level of LDL. There will be a wide range of audience who is interested in learning this result. The experiment design is solid, and the conclusion is supported by the data. No weakness can be identified in the current manuscript.

Response:

We highly appreciate this acknowledgement of our study from Reviewer #1

Reviewer #2 (Remarks to the Author):

This manuscript addresses the provocative concept that HSPGs serve as a receptor for PCSK9 in the liver and facilitate presentation of PCSK9 to the LDLR. They present data consistent with the concept that interruption of the PCSK9/HSPG interaction results in reduced PCSK9 activity and therefore could be a novel strategy for PCSK9 inhibition. The studies are interesting and fairly compelling with regard to the HSPG-binding of PCSK9.

Specific comments:

1. Do the antibodies against the PCSK9 HSPG binding site affect direct binding of PCSK9 to the LDLR in absence of HSPGs?

Response:

The reviewer raises an important point. We have now included an experiment showing that mAb 5E11 directed against the PCSK9 HSPG binding site does not prevent PCSK9 binding to the purified extracellular domain of LDLR in a cell-free assay (Supplementary Fig. 7c).

2. Does injection of WT PCSK9 affect LDLR protein in non-hepatic tissues? Is this effect altered by co-injection of heparin or suramin?

Response:

We have included new data showing that whereas PCSK9 induces LDLR degradation in the liver it does not do so in the adrenal gland (Supplementary Fig. 8).

3. The majority of the in vivo studies employed the injection of exogenous PCSK9. Do heparinase and heparin increase the hepatic LDLR protein due to inhibition of endogenous PCSK9?

Response:

We have included new experiments showing that transgenic overexpression of human heparanase results in markedly increased LDLR protein in the liver despite an increase in plasma PCSK9 (Fig. 6e-h).

4. The only experiment relying on endogenous PCSK9 is represented by Fig 6de, an experiment involving a single injection of suramin in WT and PCSK9 KO mice. The changes in LDLR and total cholesterol are very modest (and in WT mice total cholesterol primarily represents HDL cholesterol). More evidence that this pathway is relevant to endogenous PCSK9 in vivo would substantially strengthen the manuscript.

Response:

We have included new experiments showing that acute heparinase injection results in release of endogenous PCSK9 (Fig. 6c-d) and that transgenic mice overexpressing human heparanase display significantly increased plasma PCSK9 levels, liver LDLR levels and reduced plasma cholesterol (Fig. 6e-h). These data show that targeting heparan sulfate proteoglycans interferes with endogenous PCSK9 activity. We have also included data showing that the natural loss-of-function PCSK9 mutant R93C does not bind heparin (Fig. 1e). R93 is part of the PCSK9 HSPG binding site. Individuals carrying R93C display low LDL cholesterol levels and are protected from coronary artery disease but the mechanism

for this has so far been unknown (Miyake et al. 2008, Tang et al. 2015). We have elaborated on this in the discussion.

Reviewer #3 (Remarks to the Author):

NCOMMS1702733T

This manuscript describes the presentation of PCSK9 to the LDL-R by a heparan sulfate (HS) proteoglycan (PG). This is a well-conducted study and an impactful paper. The results presented in the six data figures are strongly supportive of the author's conclusions. Figure 1 clearly confirms the binding of heparin, although the x-axes on panels k,l,m should be revised to present the heparin concentrations in micrograms/ml instead of units/ml as units of anticoagulant activity are completely unrelated to the activities discussed in this manuscript.

Response:

The x-axes have now been changed in the revised manuscript and now present heparin concentrations in micrograms/ml.

Also the text suggests that PCSK9 contains a "perfect" binding site—what is a perfect binding site?

Response:

We apologize for the lack of clarity and have deleted the word "perfect".

In Figure 2 it is unclear why S-dC-38 was tested some more explanation is required here.

Response:

S-dC-38 is a phosphorothiorate oligonucleotide, which is highly anionic and known to interact with heparin binding proteins (Guvakova et al 1995, Fennewald et al. 1995). This is now explained in the results section and the citations are included.

Figure 3 is and the reinterpretation of the x-ray crystal structure is very convincing. Figure 4 suggests to this reviewer that three trisulfated disaccharide repeating units (present in the heparin I structure—this reviewer does not think it appropriate to refer to these oligos as heparins, maybe heparin oligo I would be more appropriate) bind tightly to PCSK9 but would four (or five) repeats bind even more tightly? Also why is natural heparin a weaker binder? Some more discussion on these points would be helpful.

Response:

We have changed name to "heparin oligos" throughout the text. We agree with the reviewer that additional disaccharide repeats may increase binding further and have now discussed this in the

results section. We have also discussed why natural heparin may be a weaker binder – it is a heterogeneous mixture of variable chain length and sulfation pattern.

In Figure 5 some information on the differences between rat and human PCSK9 would be useful.

Response:

We have included information on the sequence identity between rat and human PCSK9 in the figure 5 legend.

In Figure 6 why was suramin selected and not the heparin oligo I structure used?

Response:

We agree that in vivo experiments using heparin oligo I would have been optimal. Unfortunately, we only had access to a very limited amount of synthetic heparin oligo I. This was sufficient for the glycan microarray but nowhere near the amounts required for mouse experiments.

The implications of this study are profoundly important as the discussion section points out. Therefore, a better cartoon for Figure 7 is warranted. The current version of Figure 7 clearly shows capture but not presentation.

Response:

We have improved the cartoon in the new figure 8 so that it shows both capture and presentation. We have changed the figure legend accordingly.

Furthermore, the discussion of the liver HS-PG is lacking and requires a more thorough discussion. There is substantial literature on the HS-PG as a liver ApoE co-receptor as well as involved in the tropism of malaria (through binding the CS protein) and hepatitis C and dengue (through binding their envelop proteins). These interactions all involve multiple trisulfated disaccharide repeating units. Are these all using the same HS-PG? Is this HS-PG a glypican or a syndecan (your cartoon structure in Figure 7 implies an integral transmembrane protein—a syndecan)? Or is the liver simply biosynthesizing highly sulfated HS glycosaminoglycan chains on all its HS-PGs?

Response:

We thank the reviewer for bringing it to our attention that the malaria parasite, dengue virus and hepatitis C all employ highly sulfated HSPG for liver-specific targeting. We have included a paragraph in the discussion on this topic in addition to a discussion of the chemical properties of liver HSPG and their expression profile. At present, we cannot conclude on whether it is a syndecan, glypican or both. We consider it likely that both glypicans and syndecans with GAG chains encompassing the correct sulfation pattern may serve as PCSK9 receptors. However, it is possible that sorting motifs in the syndecan cytoplasmic tails may contribute to altering the trafficking route of LDLR from recycling to degradation. The cartoon indicates a single pass transmembrane receptor such as a syndecan and we now comment on this in the new figure 8 legend.

REVIEWERS' COMMENTS:

Reviewer #2 (Remarks to the Author):

The authors have addressed my comments.

In revised Figure 8, the legend should denote the red dot as "LDL" not "cholesterol"

Reviewer #2 (Remarks to the Author)

The authors have addressed my comments.

In the revised Figure 8, the legend should denote the red dot as “LDL” not “cholesterol”

Response to Reviewer 2

”Cholesterol” has now been changed to ”LDL” in the revised Figure 8.